On the enigma of Palaenigma wrangeli (Schmidt), a conulariid with a partly non-mineralized skeleton

Kröger Björn bjorn.kroger@helsinki.fi 1
Vinn Olev 2
Toom Ursula 3
Corfe Ian J. 4
Kuva Jukka 4
Zatoń Michał 5
1 Finnish Museum of Natural History, University of Helsinki , Helsinki , Finland
2 Institute of Ecology and Earth Sciences, University of Tartu , Tartiu , Estonia
3 Department of Geology, Tallinn University of Technology , Tallinn , Estonia
4 Geological Survey of Finland , Espoo , Finland
5 Institute of Earth Sciences, University of Silesia in Katowice , Sosnowiec , Poland
De Baets Kenneth
Electronic publication date: 2021 Nov 2
Publication date: 2021
Volume: 9
Electronic Location ID: e12374
Received 2021 Apr 19; Accepted 2021 Oct 3
Copyright: ©2021 Kröger et al.
Copyright year: 2021
Copyright holder: Kröger et al.
License: This is an open access article distributed under the terms of the Creative Commons Attribution License, which permits unrestricted use, distribution, reproduction and adaptation in any medium and for any purpose provided that it is properly attributed. For attribution, the original author(s), title, publication source (PeerJ) and either DOI or URL of the article must be cited.
License URL: https://creativecommons.org/licenses/by/4.0/

Keywords: Scyphozoa, Cnidaria, Ordovician, Estonia, Apatite skeleton, Tetraradiate

Funding: The Academy of Finland (Björn Kröger) No. 309422 RAMI infrastructure project No. 293109 The Estonian Research Council PRG836 This work was supported by the Academy of Finland (Björn Kröger: No. 309422; Jukka Kuva, Ian J Corfe: through RAMI infrastructure project No. 293109) and by the Estonian Research Council (Ursula Toom, Olev Vinn: grant PRG836). The funders had no role in study design, data collection and analysis, decision to publish, or preparation of the manuscript.

==============================
Palaenigma wrangeli (Schmidt) is a finger-sized fossil with a tetraradiate conical skeleton; it occurs as a rare component in fossiliferous Upper Ordovician strata of the eastern Baltic Basin and is known exclusively from north Estonia. The systematic affinities and palaeoecology of P. wrangeli remained questionable. Here, the available specimens of P. wrangeli have been reexamined using scanning electron microscopy and x-ray computed tomography (microCT). Additionally, the elemental composition of the skeletal elements has been checked using energy dispersive X-ray spectroscopy. The resulting 2D-, and 3D-scans reveal that P. wrangeli consists of an alternation of distinct calcium phosphate (apatite) lamellae and originally organic-rich inter-layers. The lamellae form four semicircular marginal pillars, which are connected by irregularly spaced transverse diaphragms. Marginally, the diaphragms and pillar lamellae are not connected to each other and thus do not form a closed periderm structure. A non-mineralized or poorly mineralized external periderm existed originally in P. wrangeli but is only rarely and fragmentary preserved. P. wrangeli often co-occurs with conulariids in fossil-rich limestone with mudstone–wackestone lithologies. Based on the new data, P. wrangeli can be best interpreted as a poorly mineralized conulariinid from an original soft bottom habitat. Here the new conulariinid family Palaenigmaidae fam. nov. is proposed as the monotypic taxon for P. wrangeli.

Introduction

The fossil Palaenigma wrangeli (Schmidt, 1874) is small, less than a small finger in diameter and no more than a couple of centimeters long. P. wrangeli has a peculiar tetraradiate symmetry with four, strange horn-like spines or pillars at each corner, and it consists of a shiny, dark-brown calcium phosphate, which cannot be overlooked on a freshly broken Ordovician limestone.

The species name refers to Wilhelm F. Baron von Wrangell (1831–1894) (son of the famous seaman Ferdinand von Wrangel), who found this fossil not far from his manor house when he was a young man only to urge twenty years later the Geologist Friedrich K. Schmidt (1832–1908) to solve its mystery. In his original description of the fossil, Schmidt (1874) reported the difficulties in finding more material. It took him two years and hours of focused searching to find another good specimen in a small quarry, where Wrangell guided him and his younger Swedish colleague Jonas. G.O. Linnarson (1841–1881). The quarry exposed the Lyckholmsche Schicht (corresponding to the Nabala and Vormsi Regional stages) and, according to Schmidt (1874), was very rich in conulariids. The dark phosphatic shell of conulariids and the skeleton of P. wrangeli stand out in the greenish-pale limestone, and if P. wrangeli were abundant, it would have been easy for the experienced and dedicated fossil hunters to find more material.

P. wrangeli is generally a rare fossil, known exclusively from Estonia, and from Pleistocene erratic blocks from the Åland Islands, Finland and Uppland, Sweden (Holm, 1893). In the palaeontological collections of Estonia only seven specimens have been accumulated until now. The specimen found by Wrangell and the four or so, original specimens collected by Schmidt and Linnarson are unfortunately lost. One specimen, probably collected by Linnarson, is in the collections of the Naturhistoriska Riksmuseet Stockholm (Sweden). All come from north Estonian light-coloured Upper Ordovician limestone, which is generally poor in skeletal intraclasts (Fig. 1).

Figure 1 Occurrences of Palaenigma wrangeli (Schmidt, 1874) in north Estonia.

(A) Map of Baltoscandia with national boundaries and capitals (black dots). (B) Map of Estonia with P. wrangeli occurrences discussed herein (yellow dots), and with outline of Late Ordovician facies belts (from Harris et al., 2004). (C) Middle –Late Ordovician Regional stages of Baltoscandia (stars mark occurrences of P. wrangeli. Hirn., Hirnantian. Map data: R Package, maps Version 3.3.0 (https://cran.r-project.org/web/packages/maps/maps.pdf).

Schmidt (1874) couldn’t solve the mystery of Palaenigma, for which he created the separate genus Tetradium, a name that was already preoccupied by another enigmatic tetraradiate fossil organism (see Walcott, 1886; Steele-Petrovich, 2009). He speculated that it could be an operculum of a conulariid. Walcott (1886, p. 224) compared it with the Cambrian calcitic polyplacophoran Mattheva Walcott. Before, Lindström (1884, p. 41), in his opus magnum on Silurian gastropods of Gotland, excluded any relation with mollusks and curiously suggested that it might be a conulariid infected by a parasitic fungus. Later, Sinclair (1952) placed Palaenigma without comment into the Conulariinae, a subfamily of the Conulariida. The conulariid affinities of P. wrangeli also appeared unquestionable for Brood (1995), who briefly described the species and interpreted it as a basal part of Conularia Sowerby (Brood, 1995). The genus, however, was not included in the review and cladistic analysis of the Conulariinae carried out by Moraes Leme & Van Iten (2008).

New finds from a small quarry in central Estonia exposing the Saunja Formation (Nabala Regional Stage) give reason to take the mysterious species under new scrutiny using modern analytical techniques. Here we describe the new material and review existing specimens available from the Estonian geoscience data platform (SARV, https://geocollections.info/), and the Naturhistorisk Riskmuseet Stockholm (NRM). SARV unites the large palaeontological collections from Estonia, and the specimens analyzed herein came from the Department of Geology at Tallinn University of Technology (GIT).

Methods

Four of the eight available specimens (see below) were investigated with a GE phoenix v—tome—x s X-ray computed tomography (micro CT) device at the Geological Survey of Finland in Espoo, Finland. The samples were imaged using an accelerating voltage of 80–100 kV and a tube current of 120–220 µA, for a tube power of 12–22 W. Tube power was kept low enough to avoid spot size—related blurring for the obtained resolutions of 12–20 µm. 0.1 mm of Cu was used as a beam filter in most scans. 2,200–2,500 angle steps were used and at each angle the detector waited for a single exposure time and then took an average over three exposures, with the single exposure time varying between 500–1,000 ms. This resulted in total scan times of 73–167 min. The obtained projections were reconstructed using GE phoenix datos—x and investigated using ThermoFisher PerGeos 2020.2.

The microstructural features of the specimens have been analyzed using a Thermo Scientific Quanta 250 analytical scanning electron microscope (SEM) housed at the Institute of Earth Sciences in Sosnowiec, Poland. The specimens have been inspected in uncoated states in low vacuum conditions using back-scattered electrons (BSE) imaging. We used BSEs because they deliver best quality images for skeletal fossils in a limestone matrix. BSE images were collected using a Directional Backscatter Detector under the following operating conditions: 13 mm working distance, low vacuum mode (40 Pa chamber pressure, water vapour atmosphere), 15 keV beam accelerating voltage, and a 200 µm aperture. Both transverse and sagittal sections of the specimens have been investigated. The elemental composition of building structures and layers have been checked using an energy dispersive X-ray spectroscopy (EDS). EDS was conducted using a Thermo Scientific Noran System 7 and an UltraDry Premium EDS detector using the same operating conditions as above. A Pathfinder EDS software was used for acquisition of point counts and counts in a microarea. EDS analyses have been performed on nine locations in different parts of the skeleton. At each location five to eleven points were analyzed in order to evaluate the variability of the results (see Article S1). SEM and micro CT images have been graphically improved by adjusting whole image Gamma and Contrast levels using Affinity Photo Version 1.9.2 graphical software.

Herein, a few descriptive terms are used (Fig. 2), which are mainly borrowed from the literature about conulariids: Periderm denotes the exoskeleton of conulariids. Carinae are broad, internal thickenings of the periderm that can be situated on the sides of the periderm or as keel-like, continuous thickenings at the corners of the periderm. In many conulariids there are multiple kinds of internal thickenings, collectively assigned by Van Iten (1992) to 11 types of internal midline (interradial) structures and two types of internal corner (perradial) structures. Septa are longitudinal walls, keels, and deep ridges in the interior of the periderm positioned at the midline. Diaphragms are horizontal truncations of the periderm well above the apex. At the position of a diaphragm the periderm tapers to an imperforate, usually adapically convex transverse wall, often also called the “apical wall” (e.g., Babcock, Feldmann & Wilson, 1987) or “schott” (e.g., Kiderlen, 1937; Van Iten, 1991).

Figure 2 Schematic illustration of the morphological features of of Palaenigma wrangeli (Schmidt, 1874).

The descriptive terms are burrowed from the literature about conulariids (see text for details).

The compilation of conulariid specimens is based on a search in the SARV database (accessed 08.04.2021) under the following link: http://geocollections.info/specimen?specimen_number_1=1specimen_number=collection_id_1=1collection_id=classification_1=2classification=taxon_1=2taxon=conulariname_geology_1=1name_geology=country_1=1country=locality_1=1locality=stratigraphy_1=11id_1=5id=depth_since_1=12depth_since=depth_to_1=13depth_to=agent_1=1agent=reference_1=1reference=original_type_1=1original_type=part_1=1part=date_taken_since_1=12date_taken_since=date_taken_to_1=13date_taken_to=dbs%5B%5D=1dbs%5B%5D=2dbs%5B%5D=3currentTable=specimenmaxSize=5page=1paginateBy=25sort=locality__locality_ensortdir=DESC.

The electronic version of this article in Portable Document Format (PDF) will represent a published work according to the International Commission on Zoological Nomenclature (ICZN), and hence the new names contained in the electronic version are effectively published under that Code from the electronic edition alone. This published work and the nomenclatural acts it contains have been registered in ZooBank, the online registration system for the ICZN. The ZooBank LSIDs (Life Science Identifiers) can be resolved and the associated information viewed through any standard web browser by appending the LSID to the prefix http://zoobank.org/. The LSID for this publication is urn:lsid:zoobank.org:pub:E466B5EF-0637-4F6F-917C-4D0D7EF41B9F. The online version of this work is archived and available from the following digital repositories: PeerJ, PubMed Central and CLOCKSS.

The Supplemental Information (Article S1, Data S1, Videos S1–S3) for this article is available at https://zenodo.org/record/5205763. Reconstructed microXCT data is deposited with the MorphoSource repository (http://www.morphosource.org) in the Palaenigma project at the following DOI addresses:

Specimen GIT 812-34: https://doi.org/10.17602/M2/M368045

Specimen NRM-Mo 153046–48: https://doi.org/10.17602/M2/M368741

Specimen GIT 655-3: https://doi.org/10.17602/M2/M368879

Specimen GIT 812-35 (includes conulariid specimen GIT 812-35-1): https://doi.org/10.17602/M2/M369289

All microXCT images and videos associated with this study are available from the CSC Fairdata-PAS service, at https://www.doi.org/10.23729/3eaf1aeb-5e0c-4704-9e18-a925688f810b.

Geological setting

All specimens of P. wrangeli described herein have been collected from localities in north Estonia, exposing Upper Ordovician strata either in natural outcrops or in drill cores (Fig. 1A). The sediments of north Estonia are tectonically nearly undisturbed and palaeogeographically represent the eastern part of the Baltic Palaeobasin of the Baltica Palaeocontinent (Männil, 1966; Jaanusson, 1979; Nestor & Einasto, 1997). During the Late Ordovician the sedimentary deposition in north Estonia was dominated by limestone and marlstone in temperate to tropical marine settings (Cocks & Torsvik, 2005; Dronov & Rozhnov, 2007). The area comprises the North Estonian Facies Belt or North Estonian Shelf which, toward the south, grades into the Livonian Basin (Jaanusson, 1979; Nestor & Einasto, 1997, Fig. 1B). The sediments of the North Estonian Shelf are predominantly neritic to shallow marine and individual sedimentary packages are locally divided by long depositional hiati and partially by erosional horizons (Raukas & Teedumäe, 1997). A well-established regional chronostratigraphic, lithostratigraphic and biostratigraphic scheme allows for high resolution correlation of the north Estonian Upper Ordovician sediments (e.g., Raukas & Teedumäe, 1997; Nõlvak, Hints & Männik, 2006; Meidla, Ainsaar & Hints, 2014, Fig. 1C).

Material

The type locality of P. wrangeli was given by Schmidt (1874) as a quarry belonging to Küti (German “Kurküll”), a manor near Viru-Jaagupi in northeastern Estonia. According to (Schmidt, 1858) the quarry was located south-west of manor house near Aruküla village (German “Arroküll”). The quarry was abandoned a long time ago and today is untraceable, its former location is indicated by the place name Lubjaahju (Estonian for Lime Kiln) (59°11′43.2″N 26°30′00.4″E). It exposed a pale-grey limestone of the Nabala and Vormsi stages (Rõõmusoks, 1966). The quarry was repeatedly visited by Schmidt because of its fossil richness (Schmidt, 1858, 1874). The abundance of conulariids was specifically mentioned and listed by Schmidt (1858, 1874) and is also documented in the SARV database by an impressive number of more than 60 conularid specimens collected from the old Küti quarry. According to Rõõmusoks (1966) the richness is mainly limited to the Vormsi Stage; he listed brachiopods (mainly Sampo hiiuensis (Öpik), Ilmarinia sinuata (Pahlen), Kiaeromena (Bekkeromena) vormsina Rõõmusoks), hyoliths, gastropods, heliolitid tabulates, rugose corals, receptaculitids, and trilobites (mainly Toxochasmops vormsiensis Rõõmusoks). One specimen of P. wrangeli from Küti are available from the collections of the NRM (NRM-Mo 153046–48)) (Figs. 3C–3D). The lithological information available from matrix of these specimens is consistent with an origin from the Kõrgessaare Formation, Vormsi Stage. The Kõrgessaare Formation consists of an argillaceous, heavily bioturbated, greenish to yellowish pale-coloured mud-wackestone (Oraspõld & Kala, 1980).

Figure 3 Specimens of Palaenigma wrangeli (Schmidt, 1874).

(A, B). Specimen GIT 575-43 from Mäemetsa Quarry, Harju County, Estonia, Nabala Stage, arrow indicates traces of midline at outer wall, photos by G. Baranov, Tallinn. (C, D). Specimen NRM-Mo 153046–48, from Küti quarry, near Viru-Jaagupi in northeastern Estonia, Vormsi Stage. (E). Specimen GIT 655-2, from Ellavere drillcore, Järva County, north-east Estonia, Vormsi? Stage. (F, G). Specimen GIT 812-34, from Sutlema quarry, west of Sutlema village, Rapla County, Estonia, Nabala Stage, photos by G. Baranov, Tallinn.

Two specimens (GIT 812-34, Figs. 3F–3G, and GIT 812-35) were collected at the Sutlema quarry, west of Sutlema village, Rapla County, central Estonia (59°10′26.28″N, 24°37′2.62″E). The active quarry exposes the Saunja Formation (Nabala Stage) and the Kõrgessaare Formation (Vormsi Stage). Both specimens came from the Saunja Formation. At Sutlema the Saunja Formation contains a rich fauna and flora, dominated by green algae (Vermiporella Stolley, Coelosphaeridium Roemer, and an unidentified delicate dendroid form), gastropods (large Murchisonia-like forms, Hormotoma insignis Eichwald) and sponges. Additionally, bivalves, brachiopods [Kiaeromena (Bekkeromena) ilmari Rõõmusoks], cephalopods, conulariids, receptaculitids, rugose corals, trilobites, stromatoporoids, and dendritic graptolites (Dictyonema sp.) occur. The rich fauna of the quarry needs a detailed taxonomic examination. The Saunja Formation is more than 10 m thick at Sutlema and consists of a bioturbated, massively bedded, light-colored mud-wackestone, typical for the Baltic Limestone Facies (Kröger et al., 2019).

Three additional specimens come from drillcores, with little information on co-occurring fauna:

Specimen GIT 655-1 was collected from Kükita 24 drillcore (58°48′18.9″N 26°56′32.5″E), c. 4 km south of Mustvee, Mustvee Parish, west of Lake Peipsi, north-east Estonia, from depth 84.35 m, Tudulinna Formation, Vormsi Stage. The faunal content of the Vormsi interval of the drillcore is remarkably rich and comprises a delicate dendroid bryozoa (Stictopora sp.), a trilobite (Isotelus sp.), a hyolithid (Dorsolinevitus vomer Holm), a conulariid, and the putative cnidarian Sphenothallus (Vinn & Kirsimäe, 2015).

Specimen GIT 655-2 (Fig. 3E) was collected from Ellavere drillcore (59°0′52.42″N, 26°1′24.89″E), c. 8 km south-east-east from Järva-Jaani, Järva County, north-east Estonia, depth 92.70 m. At the same horizon occurs a bellerophontid [Megalomphala crassa (Koken)], and a brachiopod [Cyrtonotella kuckersiana cf. kuckersiana (Wysogorski)]. The specimen GIT 655-2 occurs in a greenish - grey, bioturbated argillaceous skeletal mud- wackestone of uncertain stratigraphy, probably from Vormsi Stage.

Specimen GIT 655-3 was collected from Mustvee 2322 drillcore, 3 km west of Mustvee, a village at the shore of the Lake Peipsi, north-east Estonia (58°50′5.41″N, 26°53′19.79″E), depth 69.15 m, from an interval within the Pirgu Stage. It occurs in a greenish gray, bioturbated, nodular argillaceous limestone of the Adila Formation.

Two specimens have been detected in the collections after completion of the micro CT and SEM analyses for this review: Specimen GIT 575-43, from Mäemetsa Quarry, Harju County, Saunja Formation, Nabala Stage (Figs. 3A–3B); and specimen GIT 655-4, from Pala 70 drillcore at 143.90 m, Jõgeva County, Pirgu Stage.

Results

Morphology

The available specimens show some generalities in skeletal morphology. All specimens consist of four equidistant marginal pillars with diameters of up to three mm. The distance of the pillars increases at a constant angle of c. 13° toward a maximum preserved periderm diameter of c. 10–11 mm (Specimen GIT 812-34, Fig. 4). The four pillars are apically interconnected by irregularly spaced transverse diaphragms, which are slightly irregularly curved toward the apex of the pillars (Fig. 5). The pillars have a roughly semicircular cross section, which results from a relatively loose and irregular cone-in-cone succession of superimposed tubular shell layers exclusively on the inner side of the pillars (Figs. 5B–5C, 6C, Videos S1–S2). The centers of the outer surface of the pillars are not covered with a continuous shell layer but expose, as a quasi-cross section, the complete succession of layers (Figs. 5C–5D). This results in a longitudinally carinate appearance of the outer surface of the pillars.

Figure 4 Micro-CT images of Palaenigma wrangeli (Schmidt, 1874), specimen GIT 812-34, from Sutlema quarry, west of Sutlema village, Rapla County, Estonia, Nabala Stage.

(A) Lateral view. (B) Lateral view 90° rotated along the growth axis relative to A. (C) Lateral view parallel to two pillars. Scale applies to all figures. Tried-and-true 3D XYZ cross with x-axis (red), y-axis blue, z-axis green.

Figure 5 Micro-CT images of Palaenigma wrangeli (Schmidt, 1874).

(A, B) Specimen GIT 812-34, from Sutlema quarry, west of Sutlema village, Rapla County, Estonia, Nabala Stage. (A) Sagittal cut. (B) Transverse cut. (C, D) Specimen NRM-Mo 153046–48, from Küti quarry, near Viru-Jaagupi in northeastern Estonia, Vormsi Stage. (C) Transverse cut. (D) Sagittal cut. Note the irregular spacing of the diaphragms and the continuation of the diaphragm –pillar layers in A and D, and the open half-circle cross section shape of the pillars in C and D. Scale applies to all figures.

Figure 6 Micro-CT images of Palaenigma wrangeli (Schmidt, 1874), specimen NRM-Mo 153046–48, from Küti quarry, near Viru-Jaagupi in northeastern Estonia, Vormsi Stage.

(A) Lateral view, scale applies to A, and B. (B) Lateral view 180° rotated along the growth axis relative to A. (C) Adapical view, scale applies to C, D. (D) Adoral view. Tried-and-true 3D XYZ cross with x-axis (red), y-axis blue, z-axis green.

The diaphragms are continuations of individual pillar layers or sheets, with the oldest and apicalmost diaphragms representing the most distal, oldest pillar layers. The thickness of the diaphragms is similar to that of the laminae of the pillars, c. 10–80 µm. The shape of the diaphragms can be deeply conically curved, such as in specimen Mo 153046–48 (Fig. 6), or shallow bowl-shaped, such as in specimen GIT 655-3 (Fig. 7).

Figure 7 Micro-CT image of Palaenigma wrangeli (Schmidt, 1874), specimen GIT 655-3, from Mustvee 2322 drillcore, west of Mustvee, north-east Estonia, Pirgu Stage.

(A) Lateral view. (B) Lateral view 90° rotated along the growth axis relative to A. (C) Lateral view 180° rotated along the growth axis relative to A. Scale applies to all figures. Tried-and-true 3D XYZ cross with x-axis (red), y-axis blue, z-axis green.

The transverse shape of the diaphragms is nearly quadratically and the pillars are positioned at or near the four centers of the square margins, which would correspond to the midlines of the four periderm-faces of a conulariid (Figs. 6C, 6D).

The height of the individual cones of the pillars is more than what is preserved in the available specimens and thus exceeds 15 mm. Hence, the skeletal material accreted in form of clearly distinguishable, separate layers or sheets from the outer margins of the periderm toward its center.

The apical end of the skeleton is open, and the pillars are not in contact with each other at their apical tip. The first septum occurs at a face width of 6 mm in specimen GIT 812-34 and at a face width of 8.5 mm in specimen Mo 153046–48.

In specimen GIT 655-3 the pillars are additionally thickened by massive flange-like skeletal sheets, which merge toward the periderm center with thick diaphragms (Fig. 7). A similarly thickened pillar section is preserved in specimen Mo 153046–48 (Figs. 6A, 6B).

Notably in specimen Mo 153046–48, GIT 655-1, and GIT 655-3 skeletal fragments of thin cone shaped skeletal sheets or walls with a fragile lattice-like texture are preserved in proximity of P. wrangeli (Figs. 8A–8C). These sheets in specimen Mo 153046–48 are longitudinally bent or folded forming sharp angles and flat faces. The shape of the sheets and the lattice-like structure is similar to co-occurring conulariid periderms (Fig. 8C, Video S3). In specimen GIT 655-1, fragments of a finely transversely annulated or ribbed phosphatic sheet are preserved near the outer margin of a pillar (Figs. 9A, 9D, 9A Article S1). Additionally, in specimen GIT 575-43 (Figs. 3A–3B) a part of the periderm is preserved in which the four pillars decrease in thickness along c. 15 mm and eventually taper off toward the aperture. The four pillars are positioned at the inner surface of the impressions of a tubular poorly skeletonized wall with a quadratic cross-section. This outer wall has a maximum face width of >10 mm. Faint traces of a face-midline are preserved along the adoral part of the specimen. In well preserved portions the wall surface shows a fine lattice like pattern.

Figure 8 Micro-CT image of Palaenigma wrangeli (Schmidt, 1874) and conulariid fragments.

(A) Lateral view of specimen GIT 655-3 with unidentified skeletal debris in surrounding sediment matrix and fragment of conulariid (upper right). (B) Lateral view of specimen NRM-Mo 153046–48 with unidentified skeletal debris in surrounding sediment matrix and fragment of conulariid (upper right). (C) Lateral view of conulariid GIT 812-35-1, from Sutlema quarry, Nabala stage, note also the disc-like shadow of a crinoid ossicle in the sediment matrix. Tried-and-true 3D XYZ cross with x-axis (red), y-axis blue, z-axis green.

Figure 9 Scanning electron images of Palaenigma wrangeli (Schmidt, 1874).

(A, C) Specimen GIT 655-1, from Kükita 24 drillcore, Mustvee Parish, north-east Estonia, Vormsi Stage, showing fragment of a transversally ornamented periderm? near external surface of a pillar. (B, D–H) Specimen GIT 655-2, from Ellavere drillcore, Järva County, north-east Estonia, Vormsi? Stage. (B, D, E) Area with distinctive papillate surface and with wrinkles (arrow in E). (F–H) Details showing lamellate conch cross section with empty or filamentous interspaces (arrow in F). Note also the chimney like structures (arrow in G).

Microstructure

ESEM observations of P. wrangeli reveal that different parts of the skeleton have a similar microstructure, consisting of several distinct thin lamellae (Fig. 9). Results from EDS from more than 80 points measured at lamellae surfaces and from transverse cracks of pillars at specimens GIT 655-1 and GIT 655-2 consistently indicate fluoroapatite as the skeletal material (see Article S1) and support previous assumptions (e.g., Schmidt, 1874, p. 44). This is similar to conulariids and Sphenothallus (e.g., Holm, 1893; Brood, 1995; Vinn & Kirsimäe, 2015; Ford, Van Iten & Clark, 2016). The thicknesses of individual lamellae vary, ranging from c. 10 to 80 µm. As in Sphenothallus, the boundaries between lamellae can be more or less sharp. In several places, within a single lamella much thinner (c. 0.5 to 0.8 µm thick) laminae occur which may mark here a primary lamination. The microstructure of individual lamellae seems to be homogeneous, composed of tiny phosphate crystals. Sometimes, however, within particular solid lamellae, empty spaces may occur, which are exclusively visible in ESEM observations (Figs. 9F–9H). Such spaces have a limited extent and are filled by microcrystalline calcium phosphate. In some areas the interspaces between successive laminae contain phosphatic aggregations of thin (c. 1.5–1.8 µm in diameter), branching and diverging filaments. Some of the laminae also possess pores and empty chimney-like structures, with an inner diameter up to 4 µm (Fig. 9G).

In the distalmost part of the skeleton of the specimen GIT 655-2 an extremely thin (up to 1 µm) outermost layer occurs, on which tiny (c. 16 µm in diameter), circular bumps (papillae) occur (Figs. 9B, 9D, 9E). These structures may be isolated or associated in small groups. In some places, additionally smaller wrinkle-like structures (shrinkage features?) also occur (Fig. 9E). The wrinkled and papillate layer is covered from the inside by homogeneous skeletal layers devoid of such structures.

Discussion

Interpretation of the shell microstructure

The distinct lamellae of the skeleton of P. wrangeli partly contain fine irregular vertical perforations, chimney-like structures (Fig. 9G), and additionally in some places the lamellae-interspaces are filled with a layer of fine filamentous phosphatic aggregations (Figs. 9F–9H). The chimney-like structures may be interpreted as original pore-like anatomical structures, because these structures appear to be limited to the papillate layer and there the shell lamina are often deflected toward the perforations. If so, it can be hypothesized that the papillate thin layer in specimen GIT 655-2 represents the remnants of the inner side of an external covering periderm. However, the filled perforations in other areas of the shell are less regular, and the presence of filamentous micro-apatitic aggregations (Fig. 9H, Article S1) in some of the lamellae-interlayers can be best interpreted as a product of microbial and fungal degradation of originally organic-rich laminae (compare e.g., Størmer, 1931; Podhalańska & Nõlvak, 1995). Such an interpretation is supported by findings of Broda & Zatoń (2017), which showed that filamentous fungi or bacteria bored through a cuticle of a Devonian thylacocephalan arthropod and spread horizontally between the laminae, indicating originally present organic matter within the thylacocephalan exoskeleton. Hence, the filamentous structures found in Palaenigma may indicate the presence of originally organic layers in between the phosphatic lamellae, which were post-mortem infested by boring microbial-fungal consortia. Similar, alternating phosphatic-organic layers also occur in skeletons of conulariids (Ford & Clark, 2016) and Sphenothallus (Vinn & Kirsimäe, 2015; Vinn & Mironenko, 2020). Sphenothallus, which is interpreted as a cnidarian (e.g., Van Iten et al., 2019), probably with close affinities to conulariids (Vinn & Mironenko, 2020), has a lamellate tubular phosphatic skeleton like conulariids, but differs from the latter mainly in lacking a tetraradiate skeletal symmetry and in having a distinct clonal budding pattern (Van Iten et al., 2019).

Systematic affinities

The surrounding enigma of P. wrangeli has two aspects: the first refers to the anatomical interpretation of the skeletal structures, and the second one, which relates to the first one, refers to its systematic affinity. Both mysteries can be partly solved with the new evidence from the examinations performed herein.

The preserved skeletal parts of P. wrangeli are invariably composed of calcium phosphate (see Article S1, presumably of apatite, such as in conulariids and Sphenothallus, Vinn & Kirsimäe, 2015). The 3D-reconstruction and ESEM examination of several well-preserved specimens reveals a consistent tetraradiate symmetry of the P. wrangeli skeleton with four semicircular marginal pillars, which are connected by irregularly spaced transverse diaphragms and which form a cone-like skeleton with an angle of c. 13°. The pillars and diaphragms are formed by a cone-in-cone structure of distinct sheets, which are accreted from the outer margin of the entire structure toward the center and from the apex toward the opening of the cone. Marginally, the diaphragms are not connected to each other except at the position of the pillars and thus do not form a closed structure. Similarly, the pillar-layers are open toward its margin and end abruptly at the outer surface of the pillars, resulting in a semicircular pillar cross-section and in a peculiar longitudinally lirate relief of the external pillar surface. The abrupt ending of the skeletal sheets at the margins of the diaphragms and at the external surfaces of the pillars suggests the presence of an organic or poorly mineralized outer cover or periderm, which is not fossilized in most specimens, and which served as an attachment structure and matrix for the formation of diaphragms and the external pillar surface. Traces and fragments of such a periderm are preserved in several specimens and can be reconstructed as an outer wall with quadratic cross-section and a fine lattice-like ornament.

In summary, the skeleton of P. wrangeli exhibits characters, known in its combination only in the Conulariina: (1) skeleton composed of calcium phosphate, (2) tetraradial, slender cone with thickened longitudinal septa at midline position and transverse diaphragms, (3) skeletal sheets forming irregularly and loosely spaced cone-in-cone structures. Poorly preserved, lightly mineralized phosphatic, transversely ornamented walls could be interpreted as remains of a periderm. Therefore, P. wrangeli can be best interpreted as a conulariid with a poorly mineralized marginal periderm, phosphatic apical pillars and diaphragms. The pillars with their flat external surfaces can be best interpreted as homologue to the mineralized longitudinal septa at midline position in the Conulariina (see e.g., Ford & Clark, 2016; Moraes Leme & Van Iten, 2008).

Taking the general similarities and distinct constructional differences into account, it is evident that P. wrangeli should be placed to a separate conulariid family. Here we suggest the new family Palaenigmaidae fam. nov. urn:lsid:zoobank.org:act:D1F2ED63-5711-4882-A7E2-CA521D6AE09F for Conulariina with steeply pyramidal skeletons with a thin chitinophosphatic periderm that consist of four equidistant marginal pillars, without or with poorly biomineralized outer shell; the apical end of the skeleton is open, and the pillars are not in contact with each other at their apical tip. P. wrangeli is the only species of the Palaenigmaidae fam. nov.

Palaeoecology

In his original description, Friedrich Schmidt noticed the extraordinary abundance of co-occurring conulariids with specimens of P. wrangeli in the type locality of Küti, north-east Estonia (Schmidt, 1874). A co-occurrence of P. wrangeli with conulariids was described from Baltic Limestone boulders from Sweden (Holm, 1893). And conulariids are also relatively common in the Sutlema quarry, where two specimens of P. wrangeli have been found, as well (see above).

The compilation of conulariids in the SARV database allows for an investigation of the question whether this co-occurrence of P. wrangeli with conulariids represents a general pattern. Conulariids inhabited the eastern part of Baltica basin from the Darriwilian (Kunda Stage) onwards throughout the Silurian. They reached their Ordovician abundance climax within the Haljala Stage with 127 specimens in the collections from 18 different localities. A second abundance peak was reached during the Vormsi Stage, from which 51 specimens from seven different localities are known (Fig. 10, Data S1). Most of the known specimens of P. wrangeli, including the type specimens, are also from the Vormsi Stage. This seems to support the idea that conulariids and P. wrangeli shared general habitat preferences and /or preservation pattern. Based on the specimens available for this study, P. wrangeli and the Late Ordovician conulariids of the eastern Baltica basin occur preferentially in depositional settings within an originally extraordinarily faunal-rich, calcareous soft substrate habitat (see references in section “Material”, and Toom, Vinn & Hints, 2019; Kröger et al., 2019) for evidence of widespread calcareous soft substrate at P. wrangeli occurrences).

Figure 10 Abundance ( = frequency of occurrences) of conulariids in Estonian Ordovician strata and stratigraphic occurrence of Palaenigma wrangeli (Schmidt, 1874).

Hirn., Hirnantian; n, number. Data: downloaded from SARV at 08.04.2021 (see also Methods section and Data S5).

Neither the extreme apices of P. wrangeli, nor that of co-occurring conulariids are known. Firmly skeletonized apical holdfast structures occur in Late Ordovician conulariids and conulariid-like fossils (Kozlowski, 1968; Brood, 1995; Robson & Young, 2013, see also Sendino, Broda & Zatoń, 2017). These holdfasts are discoidal or rootlet-like, indicating differentiated conulariid attachment on hard substrate (discoids) and soft substrate (rootlets). Rootlet-like skeletal appendages are often interpreted as functioning for stabilization and attachment within soft substrate (e.g., Kozlowski, 1968; Seilacher & MacClintock, 2005). The thickened and reinforced apical septa of P. wrangeli are unknown from other conulariids. As a speculation, these structures could have served as anchors, which weighted the apices down in a muddy substrate. Elongated, stick-like conch forms, similar to that of conulariid periderms occur in mud-sticking bivalves, such as Pinna Linnaeus, which shares even more similarities with conulariids in having a subquadratic conch cross section (see e.g., Seilacher, 1984). However, more complete apical material of P. wrangeli is needed to substantiate, this hypothesis.

Conclusions

Paleaenigma wrangeli (Schmidt, 1874) is a rare fossil known from few specimens collected from Upper Ordovician limestone outcrops across northern and central Estonia and from erratic boulders in Finland and east central Sweden. The systematic affinities of the monotypic Paleaenigma were disputed. A thorough analysis of well-preserved specimens with X-ray computed tomography, scanning electron microscopy, and energy dispersive X-ray spectroscopy reveal that the skeleton of P. wrangeli is composed of distinct calcium phosphate (apatite) lamellae. The lamellae are partly porous and ornamented with distinct papillae and contain poorly mineralized interlayers. The skeleton consists of four pillars, which are connected by irregularly spaced diaphragms and which are marginally open. The diaphragms are quadratic in transverse view and the pillars are situated at the four sides of the diaphragm squares. In few specimens remains of thin, poorly preserved transversally ornamented apatitic tube-forming walls are preserved near the distal margins of the pillars. Therefore, P. wrangeli can be best interpreted as a conulariid with a poorly mineralized marginal periderm, phosphatic apical pillars at midline position, and diaphragms. The new monospecific family Palaenigmaidae fam. nov. is proposed for P. wrangeli. Conulariids often co-occur with P. wrangeli. A comparison of other conulariid occurrences in Estonia with P. wrangeli occurrences indicates that these fossils are most abundant in depositional settings within an originally extraordinarily faunal-rich, calcareous soft substrate habitat. Based on its general morphology P. wrangeli can be interpreted as a poorly mineralized conulariid with a mud-sticking original life habit.

Supplemental Information

Supplemental Information 1 X-ray computed tomography video of Palaenigma wrangeli (Schmidt, 1874), specimen GIT 655-3, from Mustvee 2322 drillcore, west of Mustvee, north-east Estonia, Pirgu Stage

Click here for additional data file.

Supplemental Information 2 X-ray computed tomography video of Palaenigma wrangeli (Schmidt, 1874), specimen NRM-Mo 153045, from Küti quarry, near Viru-Jaagupi in northeastern Estonia, Vormsi Stage

Click here for additional data file.

Supplemental Information 3 X-ray computed tomography image of conulariid GIT 812-35-1, from Sutlema quarry, Nabala stage

Click here for additional data file.

Supplemental Information 4 Results of X-ray spectroscopy (EDS) of Palaenigma wrangeli (Schmidt, 1874)

Specimen GIT 655-1 from Kükita 24 drillcore, south of Mustvee, north-east Estonia, Vormsi Stage, and specimen GIT 655-2, from Ellavere drillcore, Järva County, north-east Estonia, Nabala Stage.

Click here for additional data file.

Supplemental Information 5 Results of review of conulariid occurrences in the SARV database

Numbers are occurrence counts in localities, stages, respectively.

Click here for additional data file.

We are grateful to Mare Isakar (Tartu, Estonia) and Christian Skovsted (Stockholm, Sweden) for help with finding material and early suggestions on how to proceed with the review of the material. Anna Madison (Moscow, Russia) and Aleksey Sokolov, Vadim Glinskiy, and Galina Gataulina (St. Petersburg, Russia) helped to gather information on the (missing) type material. Gennady Baranov (Tallinn, Estonia) and Duncan Matthews (Helsinki, Finland) were supportive and important companions during field trips.

Additional Information and Declarations

Competing Interests

Author Contributions

Field Study Permissions

Data Availability

New Species Registration

The authors declare there are no competing interests.

Björn Kröger conceived and designed the experiments, analyzed the data, prepared figures and/or tables, authored or reviewed drafts of the paper, and approved the final draft.

Olev Vinn conceived and designed the experiments, analyzed the data, authored or reviewed drafts of the paper, and approved the final draft.

Ursula Toom and and Michał Zatoń conceived and designed the experiments, performed the experiments, analyzed the data, authored or reviewed drafts of the paper, and approved the final draft.

Ian J. Corfe and Jukka Kuva conceived and designed the experiments, performed the experiments, analyzed the data, prepared figures and/or tables, and approved the final draft.

The following information was supplied relating to field study approvals (i.e., approving body and any reference numbers):

No permission was needed to collect the fossil material. The material is under official curation at TalTech Fossil Collection: https://www.gbif.org/dataset/8130e5c6-f762-11e1-a439-00145eb45e9a.

The following information was supplied regarding data availability:

The reconstructed microXCT data is available at MorphoSource in the Palaenigma project:

- Specimen GIT 812-34: https://doi.org/10.17602/M2/M368045

- Specimen NRM-Mo 153046–48: https://doi.org/10.17602/M2/M368741

- Specimen GIT 655-3: https://doi.org/10.17602/M2/M368879

- Specimen GIT 812-35 (includes conulariid specimen GIT 812-35-1): https://doi.org/10.17602/M2/M369289

GIT refers to Department of Geology at Tallinn University of Technology, Ehitajate tee 5, 19086 Tallinn Estonia, and NRM refers to Naturhistorisk Riskmuseet Stockholm, Swedish Museum of Natural History P. O. Box 50007 SE-104 05 Stockholm, Sweden.

All microXCT images and videos associated with this study are available from the CSC Fairdata-PAS service: https://doi.org/10.23729/3eaf1aeb-5e0c-4704-9e18-a925688f810b.

The following information was supplied regarding the registration of a newly described species:

Publication LSID:

urn:lsid:zoobank.org:pub:E466B5EF-0637-4F6F-917C-4D0D7EF41B9F

Family name:

urn:lsid:zoobank.org:act:3E1F98CB-C7DD-458B-AC45-FE366030DC02.

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
