# Peer review of "On the enigma of Palaenigma wrangeli (Schmidt), a conulariid with a partly non-mineralized skeleton"

_PeerJ, doi:10.7717/peerj.12374_

## Round 0.1 · original submission · Major Revisions

You comprehensively investigate the enigmatic Palaeonigma with new methods which is long overdue. Your investigation lends credibility to the idea that it is a conulariid cnidarian. I look forward to seeing this study published, but some crucial aspects need to be resolved before publication:

1) Documentation of the fossils: I agree with the reviewers that pictures of the main specimens under normal light conditions would be important. I also feel it would be crucial to have schematic illustrations of the main morphological features of Palaeonigma as well as those of a conulariid highlighting similarities and differences for comparative purposes. Not all readers might be that familiar with Palaeonigma and have all the literature at hand to follow the discussion. You also rightfully mention Sphenothallus in this context but this taxon and its potential affinity with conulariids or at least scyphozoans would merit some additional discussion.

2) Phosphatic composition: I agree with reviewer 2 that demonstrating a phosphatic composition is crucial to your study (to substantiate the taxonomic assignment as well as better understand the taphonomy and original composition of Palaeonigma). To be able to achieve this, your ESEM and EDS analyses still needs some work (better documentation of experimental and operating conditions, additional points analyses). It would also be crucial to show in picture/sketches (see Point 1) on which locations particularly analyses (including also CT) were performed. The results would also be more convincing if another phosphatic fossil (particularly a conulariid) from the same locality/beds would be investigated with the same methods to demonstrate their similarities (or highlight differences) in composition/preservation. Reviewer 2 gave already quite some constructive feedback but has offered to give additional feedback if you want to get in contact with him.

3) Paleoecology: Although the reviewers largely agree with a mud-sticking habit, you need to bring more convincing arguments to support this interpretation (see comments by reviewers 1-3).

4) Microbial-fungal activity: you briefly mention microbial-fungal consortia to be likely responsible for post-mortem damage of organic layers. This argument would be more convincing if you could cite some comparable damage in conulariids and/or similar kind of post-mortem damage attributable to microbial-fungal consortia from other time-slices. I do necessarily disagree with this interpretation, but some supporting references would be helpful.

5) Data accessibility and scientific reproducibility: thank you for providing a lot of supplementary material concerning your CT analyses (sections, videos). However, it would also be essential to make available the full-resolution image stack (e.g., TIFF) in addition to the final 3D models used in study (e.g., STL) in formats that can be opened by everyone for the sake of scientific reproducibility at the latest upon publication – ideally already during review (Note that some platforms like Morphobank or Morphosource allow for an embargo or password-protection for exactly these purposes). See recommendations by Davies et al. (2017).

Please address also all other points raised by the reviews as well as my own including the ones in the annotated pdfs.

Although I advise major revisions, I think most of these aspects are fairly easy to resolve. The main reason for giving major revision relates to point 2 for which in my perception the revised manuscript would need a re-review.

·

Basic reporting

This MS uses a correct profesional English. The MS is also well structured. Regarding the addional files, there is an image on the first page of peerj-60056-S4_Article_S4 that should be improved as it is blurred.
I recommend to give the information recorded on the Material chapter in a table.
Finally, there are several issues that should be addressed before its publication. Please see the PDF.

Experimental design

Reseach well exposed with rigorous study of the specimens under CT.
The method has a detailed description.

Validity of the findings

It is a meaningul study of a conulariid species neglected for almost 150 years. It is the first time that has been revisited with new technology, giving new information about its microstructure.

Additional comments

I am very glad the authors wrote this MS. It is a well-written, needed, and useful MS of an unusual conulariid species. The authors, however, need to improve the Palaeoecology chapter as the justification they use is not completely right.

·

Basic reporting

The basic reporting of the article is well done (see comments for further details).

Experimental design

The experimental design could use some work (see comments for further details).

Validity of the findings

The validity of some findings are uncertain, or at least needs to be further elaborated on (see comments for further details).

Additional comments

Review of Kröger et al, P. wrangeli
The manuscript “On the enigma of Palaenigma wrangeli (Schmidt), a conulariid with a partly non-mineralized skeleton” sheds new light on an enigmatic fossil that has been known about for sometime, but requires more advanced analysis than traditional paleontological studies could provide. The authors use their microCT method well, illustrating the fossil in fine detail. The microstructure and morphology are the paper’s strongest attributes and make the most compelling case for the author’s conclusions. The ESEM and EDS data have some of the largest gaps and do not contribute well to the stated conclusions (even though I agree with them!). Overall this is good work and the authors have shown new details of an obscure organisms. I would recommend publication after heavy modification.
See below for further comments, ranked by importance then line.
1. EDS Data Concerns: The EDS derived data is the thorniest of issues for this manuscript. It is the weakest instrument argument in the paper, but the sole reason for it even being in the MS (showing the presence of phosphate) is critical to the inferred taxonomy of the fossil. For examples of standard presentations of EDS data I would recommend looking at any article on the subject from James Schiffbauer at the University of Missouri, USA. EDS issues from the manuscript are as follows:
1a: Operating Conditions. What software and operating conditions were used during EDS collection? What kind of BSE detector was used? Was it a ‘copper wafer’ or ‘cone’ style? Why was that detector used? Why did you choose those and why did you use BSE rather than just SEM imagery? Likely it was for image clarity using ESEM conditions, but it wasn’t stated. I was impressed by the crispness of the ESEM images!
1b: Point Counts: Why are there only 2 points analyzed? Yes, there are thousands of counts for each point, but only two points is far to few to account for errors. These errors range from matrix (or other layers) inclusion or cover, time-of-flight issues, angle of materials, differing fossil preservation, or even differing vapor concentrations from the ESEM conditions. I completely understand that rare and limited fossils, with limited surface exposure, can be difficult to image, but it seems like there is more space that can be sampled. The more sampled points, the more robust your conclusions can be! If more sampling does occur, a cross-comparison with the known outgroup (the conulariids) should be considered.
1c: Context: Where are these EDS points taken? It is not clear what specimens these EDS points come from. The sample imaging spots are too zoomed in to give the reader any context to where they are on a specimen nor are they pointed out on the imaged specimens from Fig. 7. If possible, EDS maps are a great resource to show material differences in fossils.
2. CT Figures in Space: Presenting 3D data via 2D media is ALWAYS a hassle and it takes a few goes to get it right. The CT data (which is really microCT) presented here is outstanding and the details of the fossils in figures 2-6 are readily apparent. However, there are times when it is not clear how the fossils are rotated because the reader is not as familiar with the materials as the authors (again, been there). The best way to illustrate 3D specimen movement is with the tried-and-true XYZ cross (with the different coloured arrows). The author’s have already standardized the main viewing axis that most CT imagery is viewed from, so adding the XYZ parts in shouldn’t be too difficult. It can also be added to Figure 2 even though that is in a slice/cut form (which, again, has great resolution).

3. Preservation/taphonomy: This is not a critical issue for the manuscript, but it is related to many of the other topics of importance. Are there any other phosphatic organisms from the same samples (or rocks) that can be sampled with the EDS to show a similar preservation style, or at least a consistent signal in the analytical equipment? Can the matrix they have been found in support the preservation that the manuscript concludes then?
Line comments:
Line 22: This is actually microCT and should be mentioned as such throughout the MS.
31: Non-italicized P. wrangeli. Make sure to re-check MS for correct formatting.
37: This is a great opener, though the citation gets in the way a bit.
38: The description of the fossil is great, but a reconstruction (even old vs new) is missing as well as a regular light photograph. Those figures do take up space and are not critical, but they give the readers context on this enigmatic fossil.
109: This should start a new sub-section, as this is focused on collection and specimen management more than ‘methods’ proper. In my copy of this the link to the geo-collection is all there and not a hyperlink, which could be odd formatting, but should be made sure is fixed before publication.
119: We all know what a PDF is so that’s a bit odd. The author’s don’t state that every image in the PDF is a JPG do they? If that’s wording that ICZN requires, then so be it.
150: Given how much text is written to describe these specimens, images should probably be provided. I agree with the authors that the MS doesn’t need require them, but they could be put in the Supplemental Materials.
207: Is there a way to show the degrees of the pillars in the MS images?
210: This is a commentary more than suggestion: have the authors thought of doing an ontological analysis of this material? That would be interesting to see how each section was added on and what that could say about growth in these organisms.
250: Can the empty spaces be seen on any CT slices? If so, they should be pointed out or a cross-section shown.
275: Are there any other examples of microbial-fungal (maybe even boring sponge?) post-mortem damage on conulariids or other phosphatic organisms from the Ordovician? If so, they should be compared, otherwise this is just kind of hanging out there.
280: I see what you did there. Good opening, but I had to go back to the intro to get it.
300: The EDS data needs some work to make conclusion #1
310: The proposed new family requires way more description than is provided here. Maybe the authors are working on a more formal description in another MS, but without the rest of the usual description work this could cause taxonomic trouble. The section could end well with just the sentence from line 309 “Taking the general similarities…”.
344: I mostly agree with this conclusion, but this is the weakest of the conclusions reached. There is evidence of mud/soft substrate holdfasts for conulariids, but NOT at the sites with P. wrangeli, so a correlation is going to be hard to setup. Is there any evidence of any kind (from other fossils maybe) in the sampled rocks for a soft substrate? This conclusion needs refining.

·

Basic reporting

This is an interesting manuscript that describes some very worthwhile detailed instrumental analyses of an unusual fossil species. The MS is relatively short and clear, and the historic introduction is pleasant and nicely told. The science seems to be solid, but I do have some suggestions about how the results should be communicated.

First of all, the English could be improved somewhat by an edit. Although the writing is mostly good and clear, I did find that it was clumsy and non-idiomatic in quite a few places through the text. Here are a few examples:
Line 52 - 
“it would have been easy to be found by the experienced and dedicated fossil hunters.”
Line 74 - “give an opportunity to take the mysterious species under a new scrutiny”
Line 91 - “subjected for microstructural observation”
Line 244 - “This is similar as in conulariids”

Perhaps more importantly, this paper could be far more impactful if the figures are made more accessible to the reader (more "user-friendly"). In the early part of the paper, it would be extremely helpful to have a figure showing photographs of some of the fossils under normal lighting conditions. The images produced using X-ray computed tomography etc. are very strong, and the 3-D videos are beautiful, but I found I wanted to know what these unusual fossils look like to the naked eye, and I had to do a Google search for photographs early in my reading of the paper. Such photos should be included.

In the Morphology section, it would be very useful to have a diagram showing salient structural features. Since this is not a “standard” conulariid, I found that I was a bit “at sea” reading the text and trying to relate it to the images derived from tomography and SEM. A labelled diagram would help a lot; it would also be helpful to have annotations or labels added in appropriate places on the photographs.

Late Ordovician apical structures are also documented in Robson and Young, 2013:
ROBSON, S.D. AND G.A. YOUNG. 2013. Late Ordovician conulariids from Manitoba, Canada. Journal of Paleontology, 87(5):775-785.

Experimental design

The research question is very well defined, and the authors have chosen instrumental approaches that allow them to look at the fossil's structures in considerable detail. Palaenigma wrangeli is a very unusual fossil that was not well understood, so this work does clearly address a gap in our knowledge of conulariid-like forms. I am not able to comment on the details of the instrumental work, or whether it is described in sufficient detail, as I have little experience of tomography and very limited SEM experience.

Validity of the findings

Overall, the findings seem to be sound and useful. Considering the systematic affinities of this remarkable fossil, it seems entirely reasonable that the authors have assigned it to a new family.

My one question on the findings relates to Lines 336-344. If the extreme apices are not known, how can the original life habit be determined? Either I am missing something, or I don’t understand the logic in this paragraph - the authors need to review this and try to make it clearer.

Additional comments

I have a couple of remarks, in addition to what is stated in the above sections:

Line 62 - Tetradium is generally not considered to be a coral, nowadays, and is quite possibly a rhodophyte. Please look at Steele-Petrovich, 2009: https://onlinelibrary.wiley.com/doi/abs/10.1111/j.1502-3931.2008.00146.x

Line 338 - The “1” is missing from “Brood, 1995”

---

## Round 0.2 · Minor Revisions

Thank you for addressing our suggestions. The manuscript is as good as accepted. I just found some minor points I would like to see addressed before publication.

The main points:

Number of newly studied specimens: you mention various specimens and a precise number of previously described and still available specimens but I could not find anywhere the total numbers of newly investigated specimens. You mention how the number of specimens form particular localities but not a total number. It would be helpful to have this number explicitly listed at least in the Methods and/or Material section.

Phylogenetic analysis: a phylogenetic analysis likely falls outside the scope of study but I might be worth pointing that your detailed characterization of this species brings such a study closer.

Writing style: some wordings in the introduction sound quite subjective and feel rather like a novel than a scientific publication (see also comments by the reviewer). I made some suggestions in the annotated pdf on how those could be changed to convey the same message but in a more suitable style which I would like you to consider.

Please address these as well as other points raised in the annotated pdfs provided by reviewer 1 and myself.

I am looking foreword to seeing this research on addressing the enigmas surrounding Palaenigma published.

·

Basic reporting

The text is clear and well structured, with enough bibliographical references.
The images and videos have good quality.

Experimental design

It is a rigorous invstigation and well supported.

Validity of the findings

It is the review of an exisitng species and the creation of a new family (Palaenigmaidae fam. nov.). It is important for advancing in the conulariid research.

The conclusions are well stated and correct.

Additional comments

The text has been improved regarding the previous version, including the palaecology chapter.

---

## Round 0.3 · accepted · Accept

Thank you for addressing these final changes which make the manuscript even easier to follow. I look forward to seeing it published. Please consider adding a note thanking the reviewers in the acknowledgments during the proofing phase.